# Hydration Properties of Cement with Liquefied Red Mud Neutralized by Nitric Acid

**DOI:** 10.3390/ma14102641

**Published:** 2021-05-18

**Authors:** Sukpyo Kang, Hyeju Kang, Byoungky Lee

**Affiliations:** 1Department of Architecture, Woosuk University, Jincheon 27841, Korea; ksp0404@woosuk.ac.kr; 2Department of Construction Engineering, Woosuk University, Jincheon 27841, Korea; 3COCHEMS Co., Ltd. Industrial Tools Circulating Center, 160, Daehwa-ro, Daedeok-gu, Daejeon 34368, Korea; fluolbk@naver.com

**Keywords:** red mud, liquefied red mud, red mud neutralization, cement paste, compressive strength, hydration heat

## Abstract

An increasing amount of red mud (RM) is being generated globally due to the growth in aluminum production. To avoid RM pollution, low-cost methods for effectively recycling RM are being investigated. We propose a method for recycling RM as a construction material. Liquefied RM (LRM) was neutralized by nitric acid and added to cement paste, and the hydration heat, compressive strength, and hydration products were investigated. The cement paste with neutralized LRM had a higher compressive strength than that of plain cement paste and cement paste with LRM without neutralization at 1 day of aging; this indicates that nitric acid neutralization increases the early-age strength. Furthermore, the cement paste with 10% neutralized LRM showed 28 days-compressive strength and hydration heating curves similar to the plain mixture, indicating the positive impact of LRM neutralization on the strength. It was noted that a greater quantity of portlandite was produced earlier in cement paste with neutralized LRM than in that without. Therefore, the proposed method of using RM as a concrete additive has the potential to reduce the cost and environmental impact of both construction materials and RM waste management.

## 1. Introduction

Red mud (RM) is the waste generated when bauxite is processed into alumina, a raw material for aluminum production [1,2]. In general, 0.8 to 1.5 tons of RM can be generated per ton of produced alumina [3,4]. With the rapid development of the aluminum industry, approximately 1.7 billion tons of RM is generated per year globally [5,6,7]. The pH of RM is typically 10.5–12.5 owing to the hydroxide (NaOH) added during aluminum production [7,8,9].

Red mud (RM) leads to serious environmental problems due to its mass disposal and strong alkalinity [3]. Improper treatment of RM can contaminate the soil and seriously impair the soil fertility. It can also lead to groundwater resource pollution and a significantly negative impact on living organisms [10,11]. Due to its detrimental effects on the environment, it is necessary to recycle it after proper treatment when discharging [3,12]. The properties of RM and the potential to upcycle it in various applications have been investigated in several studies [6,13]. Most of them have focused on the addition of RM to cement mortar and mechanical properties of the resulting composite as a construction material [13]. These studies typically used the powder obtained from drying RM sludge to a moisture content of approximately 30–50%. However, heating, drying, and crushing processes for RM powder production increase the manufacturing costs and energy consumption, which reduces the viability of recycling the material. Hence, we propose a method for liquefying RM sludge by a simple mixing process to promote RM recycling.

Regardless of the cement type, the compressive strength of concrete tends to decrease as the liquefied RM (LRM) content and age increase [14]. This reduced compressive strength is expected to have a negative effect on the cement hydration reaction due to the high alkali content of RM [15]. By adding 10% LRM neutralized by sulfuric acid to cement, its 28-day compressive strength can represent about 99% of that plain of the plain mixture. However, the use of nitric acid-neutralized red mud significantly lowers the initial strength of the cement paste, particularly the daily strength by up to 83% and the 3-day strength by up to 68% [16].

In most previous studies, the RM was neutralized using sulfuric acid. As LRM neutralized by nitric acid is expected to have a curing effect similar to NaNO_3_ because Na^+^ in LRM and NO_3_^−^ in nitric acid are present, we chose nitric acid to neutralize RM. Nitrates, such as calcium nitrate (Ca(NO_3_)_2_) and sodium nitrate (NaNO_3_), are often used as hardening accelerators to improve the initial strength of cement [17,18]. Sufficient quantities of sodium nitrate promote the hydration of solidified cement matrices during the pre-induction period and shorten the acceleration period [19].

Therefore, in this study, LRM was neutralized by nitric acid with the aim of recycling the RM as a construction material. After adding LRM neutralized by nitric acid to cement paste, the hydration heat, compressive strength, and hydration products were investigated, especially at an early age. The findings are expected to provide useful information regarding RM upcycling.

## 2. Materials and Methods

### 2.1. Materials

In this study, RM sludge (KC Co., Mokpo, Korea) was used as LRM and nitric acid-neutralized LRM (hereafter, LRM + N). Similar to previous studies, LRM was prepared by mixing RM sludge, water, and polycarbonate thickener at a ratio of 1:0.2:0.0036 w.r.t mass of RM sludge with a moisture content of approximately 36 wt.% [16]. First, RM sludge was mixed with water for approximately 3 min using a homomixer (K&S Co. Ltd., Seongnam, Korea), as shown in Figure 1. To improve the storage stability, a thickener was then added and stirred into the mixture for 2 min [15].

LRM + N samples were prepared by adding sufficient nitric acid (6.9–17.3 g) (60%, Samchun Pure Chemical Co., Ltd., Pyeongtaek, Korea) to 100 g of LRM to achieve a pH of 11.5. The pH was measured for 720 min, and the results are shown in Figure 2. The sample with 8.6 g of nitric acid had a final pH of 7.5, which is deemed to be in a stable neutral region. Table 1 shows the physical properties of LRM and LRM + N.

Figure 3 shows the mineral properties detected from X-ray diffraction (XRD; Rigaku, SmartLab, Tokyo, Japan) patterns of these components. A characteristic peak attributed to Na(NO_3_) was observed at 2θ = 29.4° for LRM + N due to the addition of nitric acid. The composition of LRM + N was slightly different from that of LRM. The main compounds of LRM + N were quartz, calcite, bohemite, and hematite, which are also present in LRM [14].

Ordinary Portland cement (OPC; Sungshin Cement Co., Ltd., Seoul, Korea) was used in this study. Its physical properties and chemical composition provided by the supplier are listed in Table 2.

### 2.2. Cement Paste

Table 3 shows the composition of the cement pastes used in the study. For the plain mix (hereafter, referred to as “Plain”), only cement was used with a water-to-cement ratio of 0.3. For LRM and LRMN mixes, 10 wt.% LRM and LRM + N and 20 wt.% LRM and LRM + N were added to Plain. Each sample was mixed using a mortar mixer (Heungjin testing machine Co., Gimpo, Korea) for 4 min to prepare the cement paste.

### 2.3. Methods

To measure the compressive strength, the cement paste was poured into a mold with dimensions 40 × 40 × 160 mm and cured at 20 ± 2 °C and relative humidity of 50% for 24 h. It was then demolded and cured at a temperature of 20 ± 2 °C and relative humidity of 50% for various curing times. The compressive strengths of three samples for each mix were measured after 1, 3, 7, and 28 days of curing, according to ASTM C 349 using a universal testing machine (Heungjin Testing Machine Co., Gimpo, Korea).

A multichannel microcalorimeter (TAM AIR, C80, SETARAM Company, Plan-les-Ouates, Switzerland) was used to measure the hydration heat flow. LRM and LRM + N were added to water and then mixed with the cement [20]. The hydration heat flow was measured for 72 h immediately after mixing.

To examine the hydration products, samples less than 10 mm in size and with different curing ages (1 hr, and 1, 3, 7, and 28 days) were collected and immersed in anhydrous ethyl alcohol for 1 day to stop the hydration. These samples were dried in an oven at 40 °C for 3 days. The dried samples were crushed and passed through a 200-mesh sieve for the XRD analysis (Rigaku, SmartLab, Tokyo, Japan) [16]. XRD analysis was performed using a CuKa wavelength at 45 kV and 200 mA and at 4°/min in the range of 2θ = 5–75°. Thermogravimetry–differential thermal analysis (TG-DTG) was conducted in air at temperatures of about 20–800 °C and a heating rate of 1 °C/min.

## 3. Results and Discussion

### 3.1. Compressive Strength

The compressive strengths of the cement paste with LRM with and without nitric acid neutralization, and that of the Plain mixture are shown in Figure 4. The compressive strength of Plain aged for 28 days was 61.0 MPa. The, LRM 10, LRM 20, LRMN 10, and LRMN 20 had compressive strengths of 38.0, 34.1, 61.9, and 52.9 MPa, respectively, after aging for 28 days. The addition of LRM to the cement paste reduced the compressive strength, as expected. As in previous studies, LRM is considered to have reduced compressive strength when added to cement paste because it is strong alkaline. The compressive strengths of the samples with neutralized LRM were higher than those of the LRM samples after 28 days of aging. In particular, the compressive strength of LRMN 10 was 61.9 MPa, similar to 61.0 MPa of Plain.

Figure 5 shows the compressive strength of the LRM–cement paste with and without neutralization normalized to that of Plain. After 1 day of aging, the compressive strength ratios were 79.2%, 1.8%, 118.5%, and 110.1% for LRM 10, LRM 20, LRMN 10, and LRMN 20, respectively. Hence, it was clear that the neutralization of LRM with nitric acid minimized the drop in the strength of the cement due to the addition of the LRM. In particular, the neutralized samples had higher compressive strengths than Plain after 1 day of curing. In addition, LRM 20 had a particularly low compressive strength ratio of 2%, which was consistent with the results of a previous study showing that the addition of LRM greatly reduced the early compressive strength of cement [16]. In a previous study, a cement paste with LRM neutralized by sulfuric acid had a compressive strength ratio of less than 90% after 1 day of aging [16], while our LRMN samples neutralized by nitric acid had a higher strength than the Plain sample. This is attributed to the accelerated hydration and highly developed strength in the presence of NaNO_3_, which was generated by the nitric acid neutralization of LRM, as confirmed by the XRD results in Figure 3 [19].

The compressive strength ratios after 3 days of aging were 64.7% and 19.5% for LRM 10 and LRM 20, respectively, and 75.4% and 68.3% for LRMN 10 and LRMN 20, respectively. Although the LRMN samples had higher compressive strength ratios, different trends are noted compared to the results after 1 day of aging. In particular, the compressive strength of the LRMN samples exceeded that of pure cement after 1 day of aging but decreased to approximately 80% of that after 3 days of aging. The strength of LRMN 10 tended to recover by 28 days, while that of LRMN 20 remained around 80%.

The compressive strength ratios after 28 days of aging were 62.3% and 55.9% for LRM 10 and LRM 20, respectively, and 99.1% and 87.2% for LRMN 10 and LRMN 20, respectively. Similar to above, the LRMN samples had higher compressive strengths. In particular, the compressive strength of LRMN 10 was comparable to that of Plain with a ratio of 99.4%. In contrast, the compressive strength ratios of the LRM samples were approximately 60%, showing a 40% reduction compared to that of Plain. The delay in the initial hydration due to the use of LRM without neutralization resulted in the low compressive strength after 28 days of aging. According to previous studies, RM content of 10% or less should be used in the construction industry because of its slower pozzolanic reaction than that of cement [21,22,23]. However, if LRM is neutralized, a higher ratio can be considered.

### 3.2. Hydration Heat

The hydration heat of cement can be divided into the hydration heat over time and the cumulative hydration heat. The former is used to indirectly predict the setting time of cement, while the latter is used to predict the initial compressive strength of the cement paste [24]. In this study, the hydration heat over time and accumulative hydration heat were measured and are shown in Figure 6 and Figure 7, respectively.

In Figure 6a, the hydration heat curve of the cement paste over time had two peaks at approximately 0.1–0.2 h and 10–55 h. Figure 6b shows the first peaks, which are related to the formation of ettringite [20,25]. Figure 6c shows the second peaks, related to the hydration of C_3_S and C_2_S, and formation of calcium silicate hydrate (C–S–H) and portlandite (Ca(OH)_2_) [26].

In Figure 6b, the peak of Plain occurred at approximately 0.1 hr, while those of LRM 10, LRM 20, LRMN 10, and LRMN 20 occurred at approximately 0.15, 0.21, 0.09, and 0.10 h, respectively. Compared to the peak of Plain, the peaks of the LRM samples were delayed, while those of the LRMN samples were synchronous. Hence, the neutralization of LRM can increase the hydration rate of the samples due to the enhanced formation of ettringite.

In Figure 6c, the peak of Plain occurred at approximately 15 h, while those of LRM 10, LRM 20, LRMN 10, and LRMN 20 occurred at approximately 24, 55, 11, and 12 h, respectively. Compared to the peak of Plain, the peaks of the LRM samples were delayed by 1.6–3.6 times, while those of the LRMN samples were synchronous. Hence, the addition of LRM + N to the cement paste caused the hydration of C_3_S and C_2_S and formation of C–S–H and portlandite to occur with similar kinetics to that of Plain.

Figure 7 shows that the accumulative hydration heat of Plain was 9.8 J/g, while those of LRM 10, LRM 20, LRMN 10, and LRMN 20 were 9.5, 6.4, 9.3, and 9.1 J/g, respectively. Compared to Plain, the accumulative hydration heat of LRM 10 is similar, while that of LRM 20 is lower by 34%. In contrast, the cumulative hydration heat of the LRMN samples is similar to that of Plain regardless of the content. Furthermore, the shapes of the accumulative hydration heat curves of the LRM samples had different shapes to those of Plain, while the LRMN samples had a similar shape. This is consistent with the results in Figure 5 showing that the addition of LRM + N to cement resulted in a higher initial strength than the addition of LRM.

### 3.3. XRD

Figure 8 shows the XRD results for Plain, LRM 20, and LRMN 20 samples over time. To investigate the initial hydration characteristics, the analysis was focused on the production of portlandite [27], which helps maintain the alkali pore solution concentration during cement hydration, thereby enhancing the C–S–H gel formation [28].

Immediately after Plain sample was prepared, peaks related to portlandite as the main hydration product were observed at 2θ = 18.1°, 34.1°, and 47.2°, as shown in Figure 8a. Portlandite is continuously generated in cement paste after the addition of water and its peak intensity increases with increasing curing time. In contrast, Figure 8b shows that the main peaks of portlandite occurred after 3 days of aging for the LRM 20 sample, which agrees with the results of previous studies, in which portlandite was not generated during the early ages when Na-based compounds were added to the cement paste [29]. Furthermore, Figure 8c shows that the main peaks of portlandite occurred 12 h after aging for LRMN 20, which is more than two days earlier than for the LRM 20 sample. These results show that the addition of LRM to the cement paste greatly delayed portlandite formation, while its neutralization reduced this effect, resulting in a delay of only 12 h with neutralization compared with 3 days without it.

In addition, the XRD analysis results are in agreement with the compressive strength results shown in Figure 4, the hydration results of C3S and C2S, and the formation of C–S–H and portlandite (as shown in Figure 6c). NaNO_3_ accelerates hydration [19] and promotes the development of the strength, as confirmed by the compressive strength results of the cement paste with LRM + N after 1 day of aging. Furthermore, the hydration progressed faster as portlandite started to appear after 28 days of hydration, portlandite and C–S–H promote the strength of all samples. Moreover, the addition of LRM + N was consistent with the results as previous studies, where no new phases were observed via XRD analysis when RM was added to concrete [29].

### 3.4. TG-DTG

The mass loss curves of the cementitious materials with increasing heating temperature have five main peaks: (1) evaporation of evaporable water and decomposition of C–S–H and ettringite, (2) decomposition of gibbsite, (3) decomposition of portlandite, (4) conversion of hematite, and (5) decomposition of calcite and katoite [30]. Figure 9 shows the TG curves of the cement paste with LRM with and without nitric acid neutralization. The weight reduction curve of the LRMN 20 sample was similar to that of Plain, but significantly different from that of LRM 20. This is attributed to the hydration products generated by the cement paste with neutralized LRMs, similar to those of Plain. This indicates that the negative impact of highly alkaline RM on the hydration reaction of cement paste, as observed in a previous study [16], can be somewhat overcome by nitric acid neutralization.

Figure 10 shows the TG-DTG curves of the cement paste with LRM with and without nitric acid neutralization at 1 h, 12 h, and 28 days of aging. The hydration products were analyzed using the evaporation of water and decomposition of C–S–H and ettringite at 30–200 °C; decomposition of portlandite at 390–460 °C; and decomposition of calcite and katoite at 650–750 °C. Portlandite (Ca(OH)_2_) is generated during the hydration of OPC. During the first stage of hydration, both crystalline and amorphous portlandite are present, while only the former is present in the later stage. The crystalline portlandite generated during the hydration process can be analyzed using TG-DTG analysis [31], which can accurately confirm the presence of portlandite.

After 1 h of aging (Figure 10a), the weight loss curve of LRMN 20 became similar to that of Plain, although the DTG curves differed. However, the decomposition of portlandite was notable for Plain but was not observed for the LRM and LRMN samples. This result agrees with the XRD results (Figure 7). After 12 h of aging (Figure 10b), both the weight loss and DTG curves of LRMN 20 were similar to those of Plain, where the peak at 390–460 °C is clearly noted, confirming the decomposition of portlandite. However, for LRM 20, no portlandite decomposition was observed, consistent with the XRD results (Figure 7) showing that no portlandite was generated until 3 days had passed. These results confirm that LRM + N improves the initial hydration compared to LRM. Finally, Figure 10c shows the TG-DTG curves at 28 days of aging. The DTG curve of LRMN 20 was similar to that of LRM 20, unlike the curves after 12 h of aging. However, the TG curve of LRMN 20 was closer to that of Plain than LRM 20. After 28 days of aging, the decomposition of portlandite was observed for all specimens, which agrees with the XRD results. Therefore, if LRM + N is added to the cement paste in an amount of ≤20%, the initial hydration is accelerated compared to Plain or LRM. In addition, the strength of the LRM + N paste was higher on day 28 compared to LRM.

## 4. Conclusions

In this study, LRM with a moisture content of approximately 35 wt.% was prepared by mixing RM with water to avoid the drying and crushing steps and then neutralized with nitric acid. The neutralized LRM was added to the cement paste, and the hydration properties were investigated. The following conclusions were drawn:The compressive strength of the LRMN samples, particularly after 1 day of aging, was higher than those of the LRM samples, suggesting the role of LRM + N in improving the strength of the cement paste at early ages. In addition, the compressive strength of LRMN 10 at 28 days was similar to that of Plain.The LRMN samples had hydration heat peaks closer to those of Plain than the LRM samples. In particular, the peaks related to the hydration products were significantly different for the LRMN and LRM samples.XRD peaks corresponding to portlandite were observed for the LRMN samples after 12 h of aging, while they were observed after 3 days of aging for the LRM samples.The TG curves of the LRMN samples were similar to that of Plain and the DTG peak related to portlandite decomposition became clear after 12 h of aging. However, for the LRM samples, this peak did not appear until 28 days of aging. This shows that the hydration product of the LRMN sample is similar to that of the Plain.

This study demonstrated the possibility of using waste RM as an additive in concrete to produce environmentally friendly building materials. As this study provides only preliminary results at early ages, further studies of the hydration mechanisms, long-term strength, and durability should be conducted in the future.

## Figures and Tables

**Figure 1 materials-14-02641-f001:**
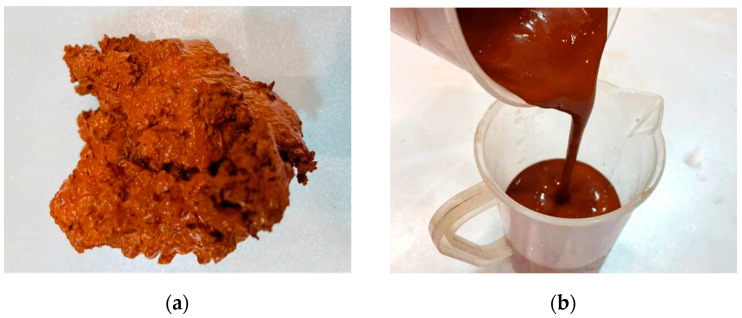
Manufacturing process of LRM: (**a**) RM sludge with a moisture content of 36 wt.%, (**b**) resulting LRM.

**Figure 2 materials-14-02641-f002:**
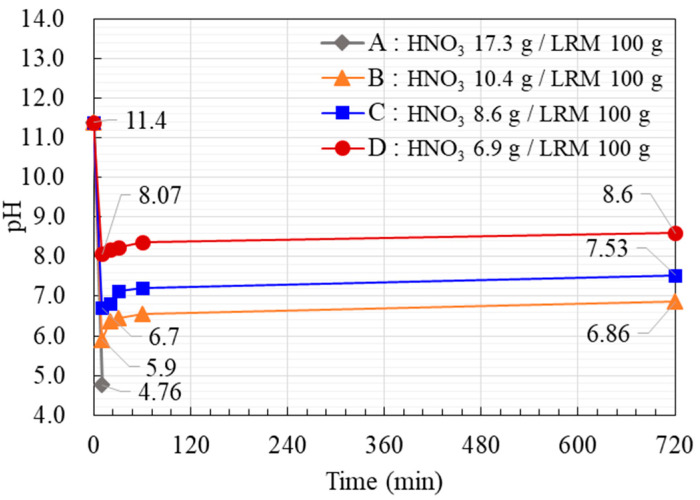
pH changes in the LRM + N samples over time.

**Figure 3 materials-14-02641-f003:**
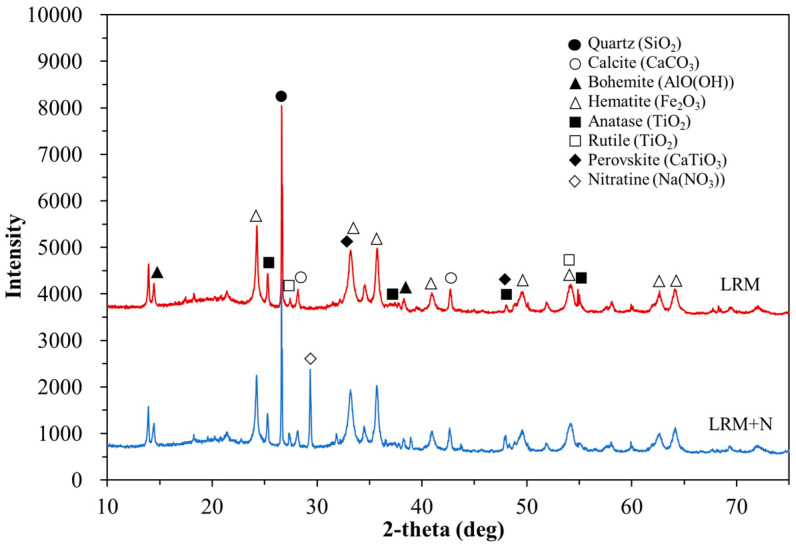
XRD patterns of LRM and LRM + N.

**Figure 4 materials-14-02641-f004:**
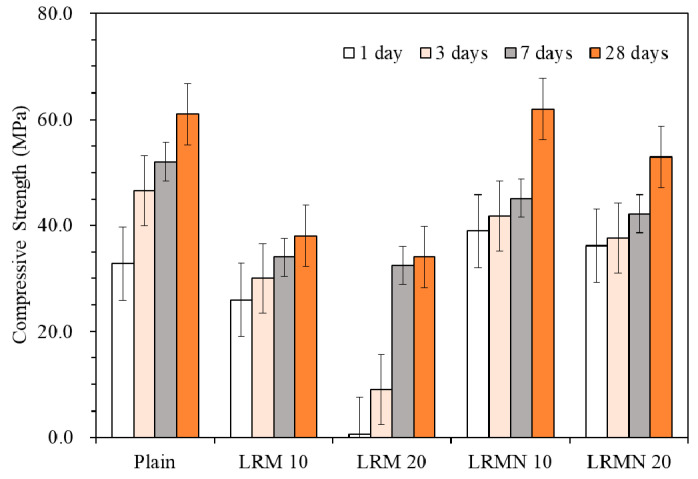
Compressive strength of the cement pastes with different compositions.

**Figure 5 materials-14-02641-f005:**
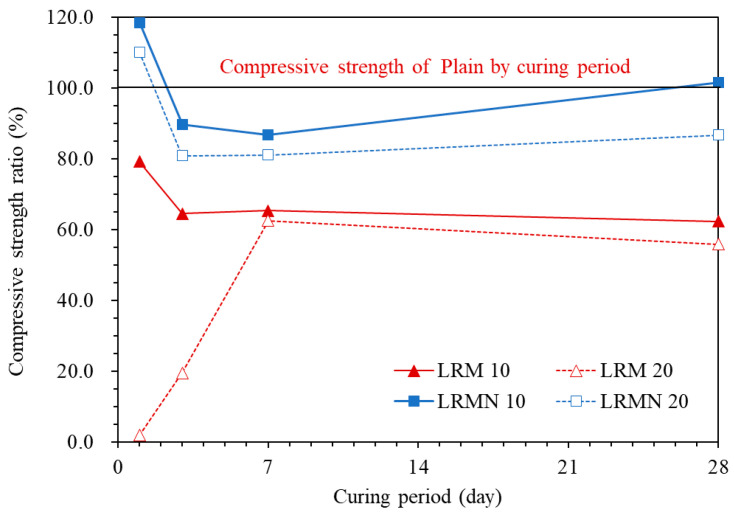
Compressive strengths of the cement pastes normalized to that of Plain as a function of the curing period.

**Figure 6 materials-14-02641-f006:**
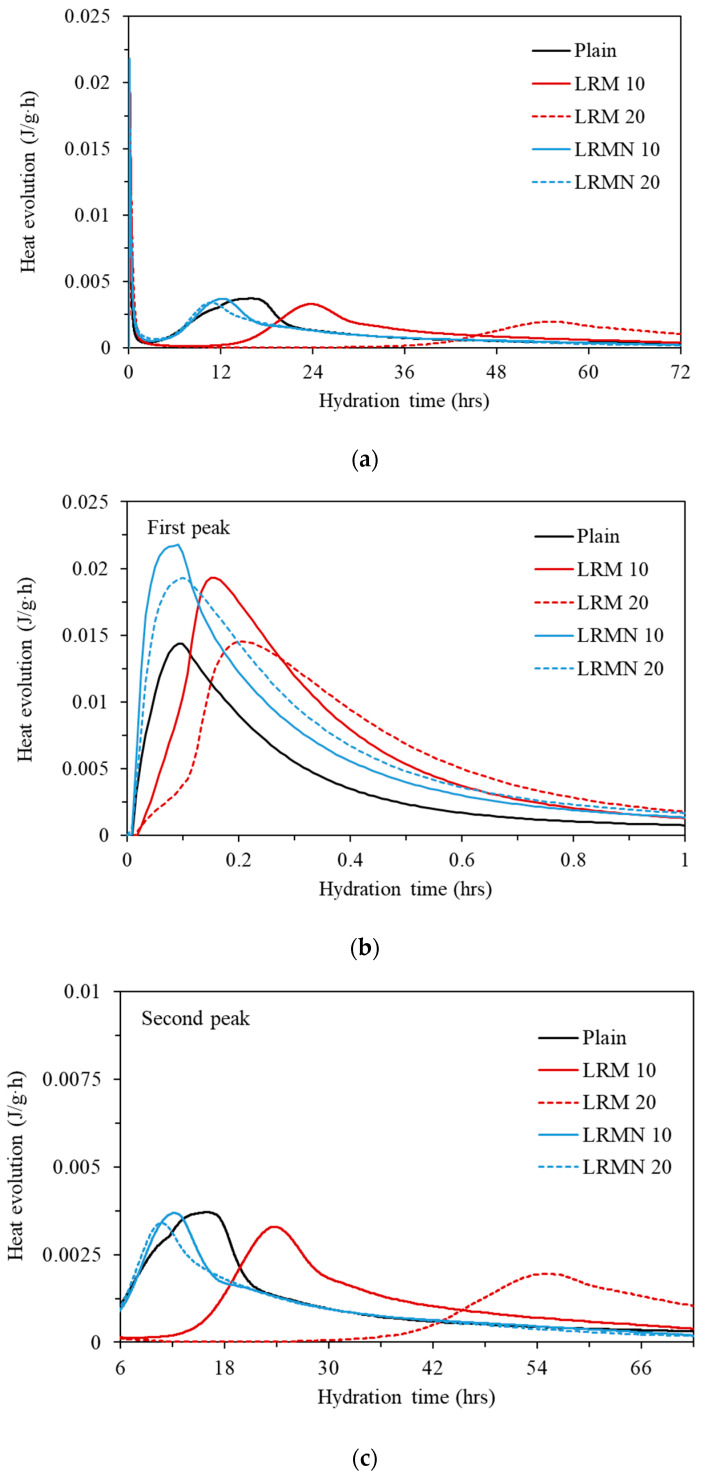
(**a**) Heat evolution rates of the various cement paste samples. Magnified view of the (**b**) first peak and (**c**) second peak in (**a**).

**Figure 7 materials-14-02641-f007:**
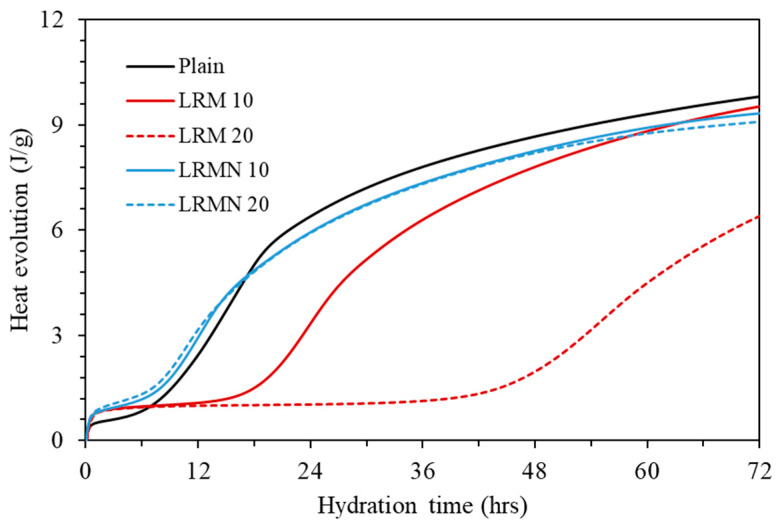
Accumulative hydration heat of the various cement pastes over time.

**Figure 8 materials-14-02641-f008:**
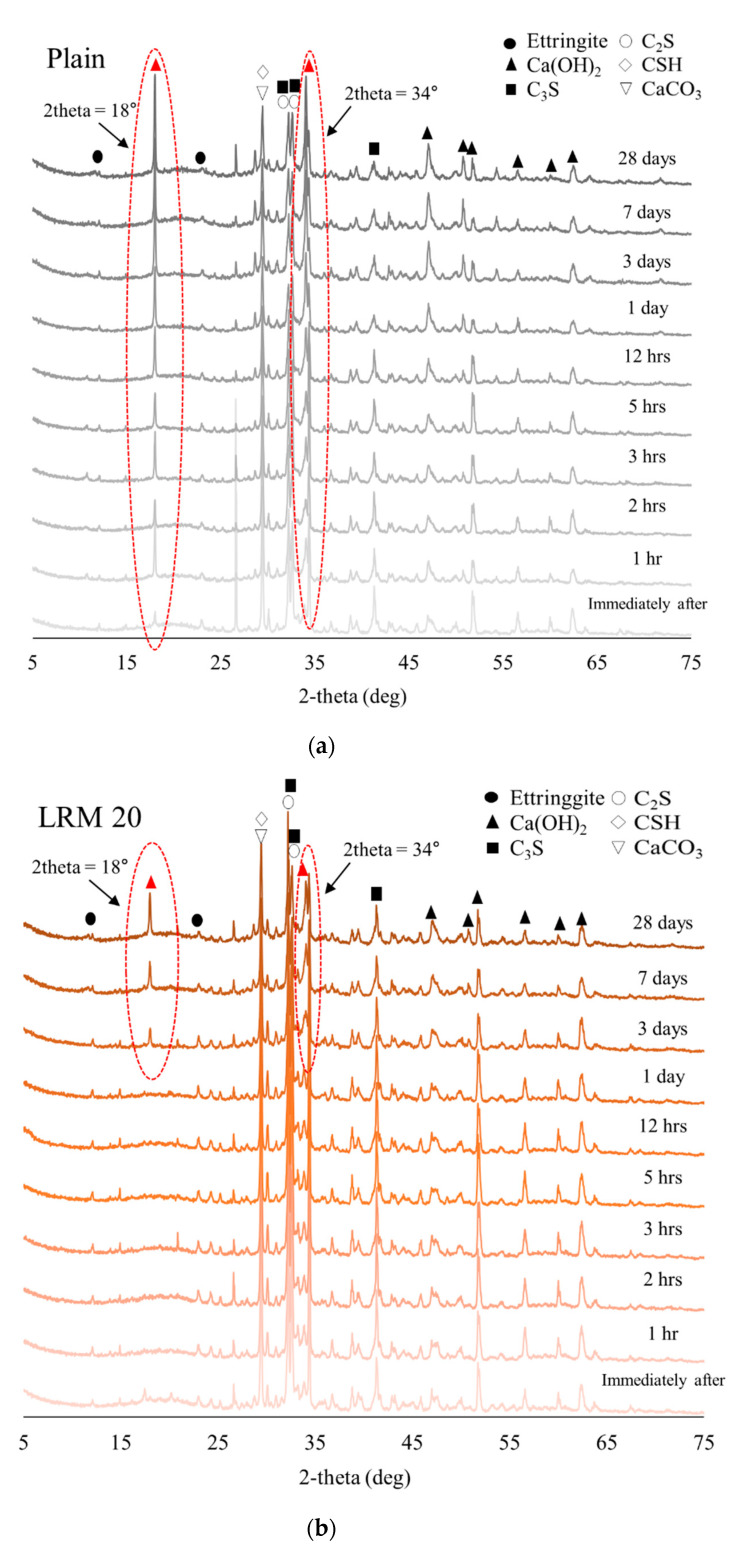
XRD spectra of (**a**) Plain, (**b**) LRM 20, and (**c**) LRMN 20.

**Figure 9 materials-14-02641-f009:**
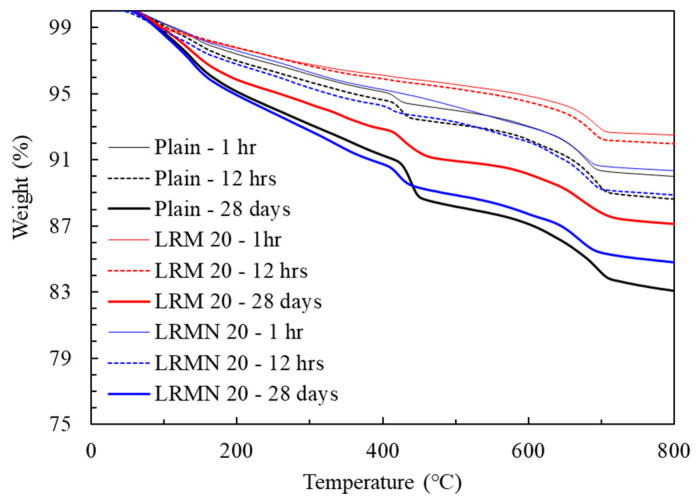
TG curves of the cement pastes.

**Figure 10 materials-14-02641-f010:**
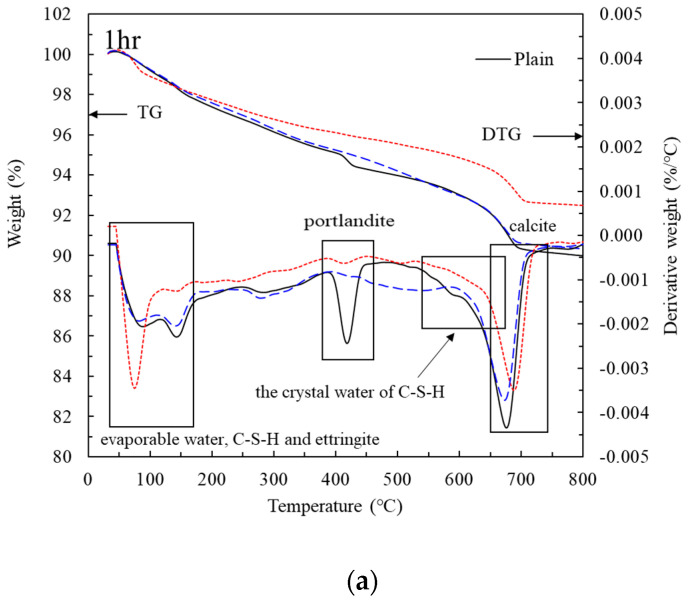
TG-DTG curves of the cement paste after aging for (**a**) 1 h, (**b**) 12 h, and (**c**) 28 days.

**Table 1 materials-14-02641-t001:** Physical properties of red mud.

Type of Red Mud	Moisture Content (%)	pH	Density (g/cm^3^)	Specific Surface Area (cm^2^/g)	Average Particle Diameter (μm)	Viscosity (cP)
LRM *	49.5	11.5	1.50	2353	2.75	42550
LRM + N **	48.7	7.5	1.50	2353	2.75	43650

* LRM: liquefied red mud. ** LRM + N: liquefied red mud + nitric acid.

**Table 2 materials-14-02641-t002:** Physical properties and chemical composition of ordinary Portland cement.

Blaine Fineness (cm^2^/g)	Setting Time		Density (g/cm^3^)	Chemical Composition (%)						
	Initial (min)	Final (h)		SiO_2_	Al_2_O_3_	Fe_2_O_3_	CaO	MgO	SO_3_	Ig-loss
3300	200	5.5	3.15	21.7	5.7	3.2	63.1	2.8	2.2	2.44

**Table 3 materials-14-02641-t003:** Mixture design.

Mixture ID	Water/Cement Ratio	Additional Component	
		LRM/Cement (wt.%)	LRM + N/Cement (wt.%)
Plain	0.3	-	-
LRM 10		10	-
LRM 20		20	-
LRMN 10		-	10
LRMN 20		-	20

## Data Availability

The data presented in this study are available on request from the corresponding author.

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
