# Peer review of "Hydration Properties of Cement with Liquefied Red Mud Neutralized by Nitric Acid"

_materials, 2021, doi:10.3390/ma14102641_

Round 1
Reviewer 1 Report
The manuscript deals with a sustainable alternative reuse of red mud, generated during the aluminium production, intended for cement applications. The topic of the paper is of high scientific and technical interest as red mud is globally considered a difficult-to-dispose polluting waste.
The paper looks quite well written and structured, however, needs some necessary improvements before being considered for publication.
The introduction should be improved with a more detailed analysis of the state of the art, explaining better why red mud could be toxic, what the general disposal procedure foresees, advantages and – moreover – disadvantages. More detailed references must also be given to current reuse, analyzing the form and the pre-treatments. That is fundamental to assess the novelty of this study in comparison to what is already known. Please, state it clearly. Finally, more attention should be given to advantages and disadvantages to nitric neutralization vs the sulfuric one.
Materials and methods should be also implemented. The ratio 1:0.2:0.0036 (line 64) should be more detailed explaining where these numbers come from. The manufacturing procedure must be structured in a more suitable way. In particular, paragraph 3.1 (lines 103-119) must be moved after line 71 as the neutralization procedure is not clear in the present form. Indeed, there is a lack of information and the reader would not understand the real manufacturing sequence. OPC must be referenced with CEM class (line 72) and in table 2 must be clarified if the characterization is experimental (done by the authors) or commercial (given by the supplier). Line 79, the produced specimen mixes are not clear (10 wt% LRM only? 20 wt% LRM+N only? It seems that also 20 wt% LRM and 10 wt% LRM+N are produced). In line 98 the XRD conditions are required.
The characterization is conversely very well presented and discussed. Particularly interesting is the hydration heat analyses. That is a good job that overall - from a scientific point of view – is suitable for an interesting publication.
Some minor inputs follow. L. 125-126 the authors should explain why it is expectable that adding red mud the compressive resistance drops and should explain why the LRMN 10 resistance is similar to that of the plain formulation (L. 128). Lines 165-168 should be reformulated as it seems the authors contradict themselves.
Finally, the manuscript needs a minor language review.
Reviewer 2 Report
The paper deals with the experimental analysis of cement liquified red mud neutralized by nitric acid in terms of the evolution of compressive strength and hydration properties. Plain cement, cement incorporating red mud sludge, and cement with liquified red mud neutralized by nitric acid were studied. For the latter two materials, two different weight percentages of liquified red mud added to cement were considered. The evolution of compressive strength was studied. XRD results, hydration heat evolution results, and thermogravimetry–differential thermal analysis results were presented. A detailed comparison was given between the different materials for each experimental test. This manuscript proposes an interesting strategy in recycling red mud in the construction industry. Although the idea of incorporating red mud in cementitious materials has been explored multiple times in the past, the use of liquified red mud neutralized by nitric acid seems to provide comparable hydration properties and compressive strength with respect to plain cement, at least for early ages.
The results are explained with details and the text is overall well written. The reviewer added a few minor comments below regarding this point.
Nevertheless, the reviewer noticed that the paper written by the same authors [13] (Kang, S.; Kang, H.; Lee, B. Effects of adding neutralized red mud on the hydration properties of cement paste. Materials 2020, 351 13, 4107) is very similar to the present manuscript, the difference being that sulfuric acid was used instead of nitric acid. Almost identical tests and figures are presented (compressive strength, calorimetry, XRD) with only minor differences in the results as compared to the ones presented in this paper. Therefore, the reviewer suggests the following:
Major revision proposed: in order to greatly improve the relevance of this study, it would be very interesting to add a section dedicated to the comparison between the performance of liquified red mud with nitric acid and with sulfuric acid. This would inform the reader about the advantages and drawbacks of each method in terms of mechanical and hydration properties (and might also include time and cost). It would also provide key points to help the interested reader in choosing one method over the other. If no important differences are found, this should also be stated.
The significance of the content and the originality would largely improve.
Minor points:
- Line 18, page 1: "Furthermore" instead of "Further".
- Line 18, page 1: "LRM had a similar [...]". Similar compared to which material?
- Lines 43-45, page 1: the authors probably meant "Due to the alkali content of RM, the cement hydration reaction has a negative impact on compressive strength [12]" ?
- Line 46, page 2: an increase by 100% with respect to plain cement means that the strength doubles from the value of strength of the plain cement. Such a result is not reported in [13].
- Line 48, page 2: "In most of the previous studies, the RM was neutralized using sulfuric acid."
- Lines 49-50, page 2: because of the presence of Na+ and NO3- ?
- Line 81, page 2: the process to make LRM+N is explained in Section Results and Discussion. It should be however introduced in the subsection Cement Paste with the other materials.
- Line 86, page 3: were the samples stored in an environmental chamber at 100% RH?
- Line 96, page 3: what were the dimensions of the specimens and how long was the immersion in alcohol? If the specimens are large enough, the immersion time should be consequent to account for the diffusion time of alcohol to attain the center of the samples.
- Line 121, page 5: "with and without" instead of "before and after".
- Line 139, page 6: the word "reduced" appears two times.
- Line 140-143, page 6: as suggested by the reviewer, a detailed comparison between LRM neutralized by nitric acid and sulfuric acid would greatly improve the impact and quality of the paper.
- Line 374-375, page 15: this citation appears two times.
Round 2
Reviewer 2 Report
No further comments.